# ON THE EXPRESSIVENESS AND SPECTRAL BIAS OF KANS

**Yixuan Wang**[1,+,*]   **Jonathan W. Siegel**[2,+]    **Ziming Liu**[3,4]    **Thomas Y. Hou**[1]
[1] California Institute of Technology
[2] Texas A&M University
[3] Massachusetts Institute of Technology
[4] The NSF Institute for Artificial Intelligence and Fundamental Interactions

## ABSTRACT

Kolmogorov-Arnold Networks (KAN) Liu et al. (2024e) were very recently proposed as a potential alternative to the prevalent architectural backbone of many deep learning models, the multi-layer perceptron (MLP). KANs have seen success in various tasks of AI for science, with their empirical efficiency and accuracy demonstrated in function regression, PDE solving, and many more scientific problems.

In this article, we revisit the comparison of KANs and MLPs, with emphasis on a theoretical perspective. On the one hand, we compare the representation and approximation capabilities of KANs and MLPs. We establish that MLPs can be represented using KANs of a comparable size. This shows that the approximation and representation capabilities of KANs are at least as good as MLPs. Conversely, we show that KANs can be represented using MLPs, but that in this representation the number of parameters increases by a factor of the KAN grid size. This suggests that KANs with a large grid size may be more efficient than MLPs at approximating certain functions. On the other hand, from the perspective of learning and optimization, we study the spectral bias of KANs compared with MLPs. We demonstrate that KANs are less biased toward low frequencies than MLPs. We highlight that the multi-level learning feature specific to KANs, i.e. grid extension of splines, improves the learning process for high-frequency components. Detailed comparisons with different choices of depth, width, and grid sizes of KANs are made, shedding some light on how to choose the hyperparameters in practice.

## 1 INTRODUCTION

Recently, in Liu et al. (2024e), a novel architecture called Kolmogorov-Arnold Networks (KANs) was proposed as a potentially more accurate and interpretable alternative to standard multi-layer perceptron (MLP) Cybenko (1989); Hornik et al. (1989). The KAN architecture leverages the Kolmogorov-Arnold representation theorem (KART) Kolmogorov (1956; 1957) to parameterize functions. KANs share the same fully connected structures as MLPs, while putting learnable activation functions on edges as opposed to a fixed activation function on nodes for MLPs. B-splines are used to parameterize the learned nonlinearity and in practice one can go beyond the two-layer construction indicated by KART. KANs can be conceptualized as a hybrid of splines and MLPs, and the combination of compositional nonlinearity with 1D splines contributes to both the accuracy and interpretability of KANs.

In this article, we study the approximation theory and spectral bias of KANs and compare them with MLPs. Specifically, we show that any MLP with the ReLU$^k$ activation function can be reparameterized as a KAN with a comparable number of parameters. This establishes that the representation and approximation power of KANs is at least as great as that of MLPs. On the other hand, we also show that any KAN (without SiLU non-linearity) can be represented using an MLP. However, the number of parameters in the MLP representation is larger by a factor proportional to the grid size of the KAN.

---

*roywang@caltech.edu
+These authors contributed equally to this work.

While we do not know if our construction is optimal in the deep case, this suggests that KANs with a large grid size may be more efficient at parameterizing certain classes of functions than MLPs. We combine these results with existing optimal approximation rates for MLPs to obtain approximation rates for KANs on Sobolev spaces.

Next, we study the spectral bias phenomenon in the training of KANs. Standard MLPs with ReLU activations (or even tanh) are known to suffer from the spectral bias Rahaman et al. (2019); Xu et al. (2019a;b), in the sense that they will fit low-frequency components first. This is in contrast to traditional iterative numerical methods like the Jacobi method which learn high frequencies first Xu et al. (2019a). Although the spectral bias acts as a regularizer which improves performance for machine learning applications Rahaman et al. (2019); Zhang et al. (2021); Poggio et al. (2018); Zhang et al. (2020); Fridovich-Keil et al. (2022), for scientific computing applications it may be necessary to learn high-frequencies as well. To alleviate the spectral bias, high-frequency information has to be encoded using methods like Fourier feature mapping Sitzmann et al. (2020); Tancik et al. (2020); Benbarka et al. (2022); Novello et al. (2024), or one needs to use nonlinear activation functions more similar to traditional methods; see for example the hat activation function Hong et al. (2022) which resembles a finite element basis. We theoretically consider a single KAN layer and show that it does not suffer from the spectral bias by analyzing gradient descent for minimizing a least squares objective. Although our analysis is necessarily highly simplified, we argue that it provides some evidence and intuition showing that KANs will have a reduced spectral bias compared with MLPs.

Finally, we study the spectral bias of KANs experimentally on a wide variety of problems, including 1D frequency fitting, fitting samples from a higher-dimensional Gaussian kernel, and solving the Poisson equation with a high-frequency solutions. Based upon the results of our experiments, KANs consistently suffer less from the spectral bias. This also helps explain why KANs are more subject to noise and overfitting, as observed in Shen et al. (2024), while grid coarsening (the opposite of grid extension) helps increase the spectral bias and reduces overfitting. Notice that the highest frequency that a single layer of KANs can learn is restricted to the number of grid points we use. The compositional structure associated with the depth also plays a role in learning the different frequencies. We study the effect of depths, widths, and grid sizes of KANs for the spectral bias, and draw conclusions about best practices when choosing KAN hyperparameters.

**Our contribution.** Our goal in this paper is to theoretically compare the KAN architecture with the commonly used MLP architecture. Our specific contributions are as follows:

- We compare the approximation and representation ability of KANs and MLPs. We show that KANs are at least as expressive as MLPs. We use this to obtain approximation rates for KANs on Sobolev spaces.
- We theoretically analyze gradient descent applied to optimize the least squares loss using a single layer KAN. Based upon this analysis, we argue that KANs, unlike MLPs, do not suffer much from a spectral bias.
- We provide numerical experiments demonstrating that KANs exhibit less of a spectral bias than MLPs on a variety of problems. This validates our theory and also provides a partial explanation for the success of KANs on problems in scientific computing.

## 2 PRIOR WORK

A theory on the approximation ability of KANs in the function class of compositionally smooth functions was proposed as KAT in Liu et al. (2024e). A convergence result independent of the dimension was obtained, leveraging the approximation theory of 1D splines. Compared with the universal approximation theorem of MLPs, KANs take advantage of the intrinsically low-dimensional compositional representation of underlying functions. This result shares an analogy to the rate in generalization error bounds of finite training samples, for a similar space studied for regression problems; see Horowitz & Mammen (2007); Kohler & Langer (2021), and also specifically for MLPs with ReLU activations Schmidt-Hieber (2020). On the other hand, for general Sobolev or Besov spaces, sharp approximation rates have been obtained for ReLU-MLPs (and more generally MLPs with most piecewise polynomial activation functions) Yarotsky (2017); Bartlett et al. (2019); Siegel (2023). These rates exhibit the curse of dimensionality, which is unavoidable due to the fact that Sobolev and Besov spaces with fixed smoothness are very large in high dimensions.

There are also subsequent works exploring other parametrizations of activation functions in the KAN formulation, including special polynomials Aghaei (2024a); Seydi (2024a); SS (2024), rational functions Aghaei (2024b), radial basis function Li (2024); Ta (2024), Fourier series Xu et al. (2024a), and wavelets Bozorgasl & Chen (2024); Seydi (2024b). Active follow-up research focuses on applying KANs to various domains, such as partial differential equations Wang et al. (2024); Shukla et al. (2024); Rigas et al. (2024) and operator learning Abueidda et al. (2024); Shukla et al. (2024); Nehma & Tiwari (2024), graphs Bresson et al. (2024); De Carlo et al. (2024); Kiamari et al. (2024); Zhang & Zhang (2024), time series Vaca-Rubio et al. (2024); Genet & Inzirillo (2024b); Xu et al. (2024b); Genet & Inzirillo (2024a), computer vision Cheon (2024b); Azam & Akhtar (2024); Li et al. (2024a); Cheon (2024a); Seydi (2024b); Bodner et al. (2024), and various scientific problems Liu et al. (2024b;c); Yang et al. (2024); Herbozo Contreras et al. (2024); Kundu et al. (2024); Li et al. (2024b); Ahmed & Sifat (2024); Liu et al. (2024a); Peng et al. (2024); Pratyush et al. (2024). In the KAN 2.0 paper Liu et al. (2024d), multiplication was introduced as a built-in modularity of KANs and the connection between KANs and scientific problems was further established in terms of identifying important features, modular structures, and symbolic formulas. In this article, we focus on the original B-spline formulation, one of the reasons being its alignment with continual learning and adaptive learning; see also Rigas et al. (2024).

The spectral bias of MLPs has been studied both experimentally and theoretically in a variety of works, see for instance Rahaman et al. (2019); Zhang et al. (2021); Xu et al. (2019a;b); Hong et al. (2022); Poggio et al. (2018); Cai & Xu (2019); Basri et al. (2020); Fridovich-Keil et al. (2022); Zhang et al. (2023); Ronen et al. (2019) and the references therein. This phenomenon is proposed as explaining the regularizing effect of stochastic gradient descent Rahaman et al. (2019); Fridovich-Keil et al. (2022). In Wang et al. (2024) an empirical analysis of the eigenvalues of the Neural Tangent Kernel Jacot et al. (2018) matrix is performed, which hints at the better performance of KANs for fitting high frequencies. However, the spectral bias of finite width KANs has not yet been considered, and it is this gap that we aim to close in this work.

# 3 REPRESENTATION AND APPROXIMATION

In this section, we study how to represent ReLU$^k$ MLPs using KANs with degree $k$ splines and vice-versa. Using this, we establish approximation rates for KANs from the corresponding results for MLPs. The measure of complexity that we consider in this section is the number of parameters of the KAN or MLP model. We remark that the number of parameters may not always be the most useful measure of complexity, especially considering that we are comparing different architectures, but it is generally indicative of the efficiency of the model. We leave the problem of comparing KANs and MLPs using different measures of complexity to future work.

## 3.1 REVIEW OF THE KAN ARCHITECTURE

We recall the following definitions of the Kolmogorov-Arnold Network (KAN) architecture introduced in Liu et al. (2024e). We refer to Liu et al. (2024e) for a thorough treatment of all details mentioned here.

**KAN architecture** The Kolmogorov-Arnold representation theorem (KART) states that any multivariate continuous function on a bounded domain can be represented as a finite composition of univariate continuous functions and addition. Specifically for a continuous $f : [0, 1]^n \to \mathbb{R}$, there exists continuous 1D functions $\phi_{q,p}, \Phi_q$ such that (see for instance Braun & Griebel (2009))

$$f(\mathbf{x}) = f(x_1, \cdots, x_n) = \sum_{q=1}^{2n+1} \Phi_q \left( \sum_{p=1}^{n} \phi_{q,p}(x_p) \right). \tag{1}$$

Inspired by KART corresponding to a two-layer neural network representation, the authors in Liu et al. (2024e) unified the inner and outer functions via the proposed KAN layers and generalized the idea of composition to arbitrary depths. A KAN can be represented by an integer array $[n_0, n_1, \cdots, n_L]$. Here $L$ is the number of layers and $n_0$ and $n_L$ are input and output dimensions respectively. $n_l$ denotes the number of neurons in the $l$-th neuron layer. Usually, we choose $n_1 = \cdots = n_{L-1} = W$

as the width and refer to $L$ as the depth of the KAN. Layer $l$ maps a vector $\mathbf{x}_l \in \mathbb{R}^{n_l}$ to $\mathbf{x}_{l+1} \in \mathbb{R}^{n_{l+1}}$, via nonlinear activations and additions, in the matrix form as

$$\mathbf{x}_{l+1} = \underbrace{\begin{pmatrix} \phi_{l,1,1}(\cdot) & \phi_{l,2,1}(\cdot) & \cdots & \phi_{l,n_l,1}(\cdot) \\ \phi_{l,1,2}(\cdot) & \phi_{l,2,2}(\cdot) & \cdots & \phi_{l,n_l,2}(\cdot) \\ \vdots & \vdots & & \vdots \\ \phi_{l,1,n_{l+1}}(\cdot) & \phi_{l,2,n_{l+1}}(\cdot) & \cdots & \phi_{l,n_l,n_{l+1}}(\cdot) \end{pmatrix}}_{\boldsymbol{\Phi}_l} \mathbf{x}_l. \tag{2}$$

The overall network can be written as a composition of $L$ KAN layers as

$$\mathrm{KAN}(\mathbf{x}) = (\boldsymbol{\Phi}_{L-1} \circ \cdots \circ \boldsymbol{\Phi}_1 \circ \boldsymbol{\Phi}_0)\mathbf{x}. \tag{3}$$

Each activation function $\phi = \phi_{l,i,j}$ in (2) is parametrized by a linear combination of $k$-th order B-splines and a SiLu nonlinearity. Namely

$$\phi(x) = w_b x/(1 + e^{-x}) + \sum_{i=0}^{G+k-1} c_i B_i(x), \tag{4}$$

where $G$ is the grid size, $B_i$ are spline basis and $c_i$, $w_b$ are trainable coefficients. The ranges of the grid points of the B-splines are updated on the fly, based on the range of the output of the previous layer. For $\phi$ on a bounded interval $[a, b]$, consider the uniform grid $\{t_0 = a, t_1, t_2, \cdots, t_G = b\}$, uniformly extended to $\{t_{-k}, \cdots, t_{-1}, t_0, \cdots, t_G, t_{G+1}, \cdots, t_{G+k}\}$. On this grid there are $G + k$ B-spline basis functions, with the $i^{\text{th}}$ B-spline $B_i(x)$ being non-zero only on $[t_{-k+i}, t_{i+1}]$.

**Grid extension** A very important structure of KANs proposed in Liu et al. (2024e) is the grid extension technique, inheriting the multi-level fine-graining from splines. One can first train KANs with fewer parameters to a desired accuracy, before making the spline grids finer with initialization from the coarser grid and continuing the training procedure. Grid extensions for all splines in a KAN can be performed independently and adaptively. We highlight that this technique is specific to the choice of splines as parameterizations of activation functions, and is particularly useful in the training process; see for example the discussion of spectral bias in the next section.

**Approximation theory, KAT** We recollect the approximation theory established for compositionally smooth functions in Liu et al. (2024e).

**Theorem 3.1.** *Let $\mathbf{x} = (x_1, x_2, \cdots, x_n)$. Suppose that a function $f(\mathbf{x})$ admits a representation*

$$f = (\boldsymbol{\Phi}_{L-1} \circ \boldsymbol{\Phi}_{L-2} \circ \cdots \circ \boldsymbol{\Phi}_1 \circ \boldsymbol{\Phi}_0)\mathbf{x}, \tag{5}$$

*as in (3), where each one of the $\Phi_{l,i,j}$ is $(k+1)$-times continuously differentiable. Then there exists a constant $C$ depending on $f$ and its representation, such that we have the following approximation bound in terms of the grid size $G$: there exist $k$-th order B-spline functions $\Phi_{l,i,j}^G$ such that for any $0 \le m \le k$, we have the bound for the $C^m$-norm*

$$\|f - (\boldsymbol{\Phi}_{L-1}^G \circ \boldsymbol{\Phi}_{L-2}^G \circ \cdots \circ \boldsymbol{\Phi}_1^G \circ \boldsymbol{\Phi}_0^G)\mathbf{x}\|_{C^m} \le CG^{-k-1+m}. \tag{6}$$

### 3.2 Reparametrization of KANs and MLPs

Next, we compare the approximation capabilities of KANs and MLPs. In particular, we have the following result showing that any ReLU$^k$ MLP can be exactly represented using a KAN with degree $k$ splines which is only slightly larger. We remark that this includes the case where $k = 1$, i.e. the popular and standard ReLU activation function.

**Theorem 3.2.** *Let $\Omega \subset \mathbb{R}^d$ be a bounded domain. Suppose that a function $f : \mathbb{R}^d \to \mathbb{R}$ can be represented by an MLP with width $W \ge 1$, depth $L \ge 1$, and activation function $\sigma_k = \max(0, x)^k$ for $k \ge 1$. Then there exists a KAN $g$ with width $W$, depth at most $2L$, and grid size $G = 2$ with $k$-th order B-spline functions such that*

$$g(x) = f(x) \tag{7}$$

*for all $x \in \Omega$.*

On the other hand, we may ask whether every KAN can also be expressed using an MLP of a comparable size. If the functions $\phi_{l,i,j}$ in the KAN representation have weight $w_b \neq 0$ in (3), then it is clear that we can not represent the resulting KAN using an MLP with activation $\sigma_k$, since the smooth function $f(x) = x/(1 + e^{-x})$ is not a polynomial. Essentially, the SiLU nonlinearity cannot be captured by a ReLU$^k$ MLP. However, if this nonlinearity is not present, then a KAN can be represented using a ReLU$^k$ MLP.

**Theorem 3.3.** *Suppose that a function $f : [0,1]^d \to \mathbb{R}$ can be represented by a KAN with width $W \geq 1$, depth $L \geq 1$, and grid size $G$ with $k$-th order B-spline functions. Assume also that the weight $w_b = 0$ in (3) for all of the activations $\phi_{l,i,j}$ appearing in the KAN. Then there exists an MLP with width $(G + 2k + 1)W^2$, depth at most $2L$, and activation function $\sigma_k = \max(0, x)^k$ which represents $f$.*

We remark that the width in this result scales like $O(GW^2)$ so that the number of parameters in the MLP scales like $O(G^2W^4L)$, while the number of parameters in the KAN scales like $O(GW^2L)$. This indicates that KANs with a large number of breakpoints at each node may be more efficient in expressing certain functions than MLPs. When the grids of the spline basis are different across each neuron pairs, which is natural after grid update in training, then there are $(G + 2k + 1)W^2$ different break points of the splines for each KAN layer. If we restrict ourselves to using only 1 hidden layer of ReLU$^k$ for each KAN layer, as in the construction of the proof in appendix C, then by He et al. (2018) it follows that the width of ReLU$^k$ MLP has to be at least $(G + 2k + 1)W^2$ and the theorem is sharp. However, we don't know whether the preceding theorem is sharp in the deep case.

Using these results, we can draw conclusions about the approximation capabilities of KANs by leveraging existing results about MLPs (see for instance Barron (1993); Leshno et al. (1993); Klusowski & Barron (2018); Siegel & Xu (2020; 2022); Hon & Yang (2022); Lu et al. (2021); Shen et al. (2022); Yarotsky (2018); Siegel (2023); Yang et al. (2023); Yang & Lu (2023)). For example, we have the following result giving optimal approximation rates for very deep KANs on Sobolev spaces (see for instance Adams & Fournier (2003) for the background on Sobolev spaces).

**Corollary 3.4.** *Let $\Omega \subset \mathbb{R}^d$ be a bounded domain with smooth boundary, $s > 0$ and $1 \leq p, q \leq \infty$ be such that $1/q - 1/p < s/d$. This guarantees that the compact Sobolev embedding $W^s(L_q(\Omega)) \subset\subset L_p(\Omega)$ holds.*

*Let $W_0 := W_0(d)$ be a fixed width (depending upon the input dimension $d$). Then for any $f \in W^s(L_q(\Omega))$ and any $L \geq 1$, there exists a KAN $g$ with width $W_0$, depth $L$, and grid size $G = 2$ with $k$-th order B-spline functions such that*

$$\|f - g\|_{L_p(\Omega)} \leq CL^{-2s/d}, \tag{8}$$

*where $C$ is a constant independent of $L$.*

This result, which follows immediately from Theorem 3.2 and the approximation rates for ReLU (and more generally piecewise polynomial) neural networks derived in Siegel (2023), shows that very deep KANs attain an exceptionally good approximation rate on Sobolev spaces. In particular, in terms of the number of parameters $P$ they attain an approximation rate of $O(P^{-2s/d})$, while a classical (even non-linear) method of approximation can only attain a rate of $O(P^{-s/d})$ DeVore (1998). This phenomenon, which is often called superconvergence DeVore et al. (2021), also occurs for very deep ReLU$^k$ networks. However, it comes at the cost of parameters which are not encodable using a fixed number of bits and thus is not practically realizable Yarotsky & Zhevnerchuk (2020); Siegel (2023).

**Remark 3.5.** Compared with the approximation theory KAT in Theorem 3.1 stated for compositionally smooth functions where the curse of dimensionality does not appear in the convergence rate, Corollary 3.4 is stated for a larger class of Sobolev functions.

## 4 SPECTRAL BIAS

In this section, we study the spectral bias of KANs and compare it to that of MLPs. The spectral bias, or frequency principle, refers to the observation that neural networks trained with gradient descent tend to be biased toward lower frequencies, i.e. that lower frequencies are learned first. This phenomenon has been well-documented and studied for MLPs, see for instance Rahaman et al. (2019); Zhang et al. (2021); Xu et al. (2019a;b); Hong et al. (2022); Poggio et al. (2018); Cai & Xu

(2019); Basri et al. (2020); Fridovich-Keil et al. (2022); Zhang et al. (2023), and it is our purpose here to develop an analogous theory for KANs. We remark that although the spectral bias is considered a form of regularization which is desirable for machine learning tasks Rahaman et al. (2019); Zhang et al. (2021); Poggio et al. (2018); Zhang et al. (2020); Fridovich-Keil et al. (2022), for scientific computing applications it is typically important to capture all frequencies and so the spectral bias may negatively affect the performance of neural networks for such applications Wang et al. (2022); Rathore et al. (2024); Cai & Xu (2019); Hong et al. (2022).

### 4.1 SPECTRAL BIAS THEORY FOR SHALLOW KANs

We consider the spectral bias properties of KANs with a single layer. This theory is very similar to the theory developed in Hong et al. (2022); Zhang et al. (2023); Ronen et al. (2019) for the spectral bias of single hidden layer MLPs. The key observation is that a single layer KAN is a linear model. In particular, we see that if $L = 1$ then the KAN applied to an input $\mathbf{x} \in \mathbb{R}^d$ is (for simplicity, we consider the case of a KAN without the SiLu non-linearity)

$$\text{KAN}(\mathbf{x}, \theta)_i = \sum_{j=1}^{d} \sum_{l=1}^{G+k-1} c_{ijl} B_l(\mathbf{x}_j),$$ (9)

where $\theta = \{c_{ijl}\}$ are the parameters of the KAN. Here $i = 1, ..., d'$ where $d'$ is the dimension of the output, $j = 1, ..., d$, and $l = 1, ..., G + k - 1$. Note that the only parameter here is the grid size $G$, since the width is determined by the input and output dimensions.

Based upon this, we can analyze least squares fitting with shallow KANs. In particular, let $\Omega = [-1, 1]^d$ be the (symmetric) unit cube in $\mathbb{R}^d$, let $f : \Omega \rightarrow \mathbb{R}^{d'}$ be a target function we are trying to learn, and consider the (continuous) least squares regression loss

$$L(\theta) = \int_{\Omega} \|f(\mathbf{x}) - \text{KAN}(\mathbf{x}, \theta)\|^2 d\mathbf{x}.$$ (10)

Due to the representation (9), this loss function is quadratic in the parameters $\theta$. Let $M$ denote the corresponding Hessian matrix, i.e. so that

$$L(\theta) = (1/2)\theta^T M \theta + b^T \theta.$$

This Hessian matrix (indexed by $i, j, l$) is given by

$$M_{(i,j,l),(i',j',l')} = \begin{cases} \int_{\Omega} B_l(\mathbf{x}_j) B_{l'}(\mathbf{x}_{j'}) d\mathbf{x} & i = i' \\ 0 & i \neq i'. \end{cases}$$ (11)

The convergence of gradient descent on the least squares regression is determined by the eigen decomposition of the Hessian matrix $M$ which is estimated in the following theorem. The theorem is a generalization of the fact that the Gram matrix of the B-spline basis is well conditioned (see for instance DeVore & Lorentz (1993), Theorem 4.2 in Chapter 5).

**Theorem 4.1.** *Given a single hidden layer KAN with grid size $G$, degree $k$ B-splines, input dimension $d$ and output dimension $d'$, let $M$ denote the Hessian matrix defined in* (11) *corresponding to the least squares fitting problem* (10). *Then the eigenvalues $0 \leq \lambda_1(M) \leq \cdots \leq \lambda_N(M)$ (here $N = (G + k - 1)dd'$) satisfy*

$$\frac{\lambda_N(M)}{\lambda_{d'(d-1)+1}(M)} \leq Cd$$ (12)

*for a constant $C$ depending only on the spline degree $k$.*

Spectral bias refers to the fact that lower frequencies converge much faster in training, which is related to the ill-condition property of the Hessian matrix in shallow models. Theorem 4.1 shows that away from $d'(d-1)$ eigenvectors the matrix $M$ is well conditioned. This means that gradient descent will converge at the same rate in all directions orthogonal to these $d'(d-1)$ eigenvectors. Note that since the number of eigenvectors we must remove is independent of the grid size $G$, we expect that when $G$ is relatively large most components of the KAN will converge at roughly the same rate toward the solution. Thus the KAN with a large number of grid points will not exhibit the same spectral bias toward low frequencies seen by MLPs.

**Remark 4.2.** We remark that in constrast the Hessian associated with a two layer ReLU MLP with width $n$ has a condition number which scales like $n^4$ Hong et al. (2022), which explains the strong spectral bias exhibited by ReLU MLPs. On the other hand, as established in Remark 14 in Zhang et al. (2023), ReLU$^k$ MLP with width $n$ has a condition number at the order of $n^{2+2k}$. One can also observe in Sec 4.3 that they perform even worse for high-frequency tasks.

**Remark 4.3.** We remark that in Wang et al. (2024) a spectral bias analysis for infinite width KANs is performed empirically using an analysis of the associated neural tangent kernel for specific tasks. Theorem 4.1 complements this work by giving an analysis in the finite width regime as well.

**Remark 4.4.** We note that the $d'(d-1)$ eigenvectors which must be excluded is not an artifact of the proof. In fact, this is due to the fact that the KAN parametrization is not unique. Indeed, the constant function $f(x) = 1$ can be parameterized in $d$ different ways by using the B-splines in each of the $d$ different directions. This ambiguity gives rise to directions in parameter space where the function parameterized by the KAN doesn't change and this results in degenerate eigenvectors of the matrix $M$.

The analysis given is necessarily highly simplified and heuristic. In particular, we only analyze a single layer of the KAN network and consider the continuous least squares loss. Nonetheless, we argue that it gives an explanation for why we would expect KANs to have a significantly different spectral bias than MLPs, and in particular why we expect that they learn all frequencies roughly similarly. In the remainder of this section, we experimentally test this hypothesis and compare the spectral bias of KANs with MLPs on a variety of simple problems. We implement these numerical experiments using the pykan package version 0.2.5.

## 4.2 1D WAVES OF DIFFERENT FREQUENCIES

In the first example, we take the same setting as in Rahaman et al. (2019) and study the regression of a linear combination of waves of different frequencies. Consider the function prescribed as

$$f(x) = \sum A_i \sin\left(2\pi k_i z + \varphi_i\right), \quad k = (5, 10, \cdots, 45, 50).$$

The phases $\varphi_i$ are uniformly sampled from $[0, 2\pi]$ and we consider two cases of amplitudes: one with equal amplitude $A_i = 1$ and another with increasing amplitude $A_i = 0.1i$. We use a neural network, either ReLU MLP or KAN, to regress $f$ sampled at 200 uniformly spaced points in $[0, 1]$, with full batch ADAM iteration as the optimizer with a learning rate of 0.0003. For MLPs, we train with 80000 iterations as in Rahaman et al. (2019); for KANs, we only train with 8000 iterations. Normalized magnitudes of discrete Fourier transform at frequencies $k_i$ are computed as $|\tilde{f}_{k_i}/A_i|$ and averaged over 10 runs of different phases.

We plot the evolution of $|\tilde{f}_{k_i}/A_i|$ during training across all frequencies; see Figure 1 and Figure 7 in the appendix B for comparisons of MLPs and KANs with different sizes for equal and increasing amplitudes respectively. KANs suffer significantly less than MLPs from spectral biases. Once the size of KANs, especially the grid size and depth is large enough, KANs almost learn all frequencies at the same time, while even very deep and wide MLPs still have difficulties learning higher frequencies, even with 10x epochs!

## 4.3 GAUSSIAN RANDOM FIELD

In this example, we consider fitting functions sampled from a Gaussian random field. The target function $f$ is sampled from a $d$-dimensional Gaussian random field with mean zero and covariance $\exp(-|x-y|^2/(2\sigma^2))$. Here small $\sigma$ corresponds to rough functions and large $\sigma$ corresponds to smooth functions.

To approximate the Gaussian random field, we sample $f$ using the KL expansion Karhunen (1947). We sample $N = 5000$ points uniformly from $[-1, 1]^d$ and calculate the (empirical) covariance matrix $K$. Then we truncate its first $m < N$ eigenpairs $\lambda_i, \phi_i$, with the cutoff threshold $\lambda_{m+1} < 0.1\lambda_1 \leq \lambda_m$ and sample $f$ approximately via

$$f = \sum_{i \leq m} \lambda_i \xi_i \phi_i,$$

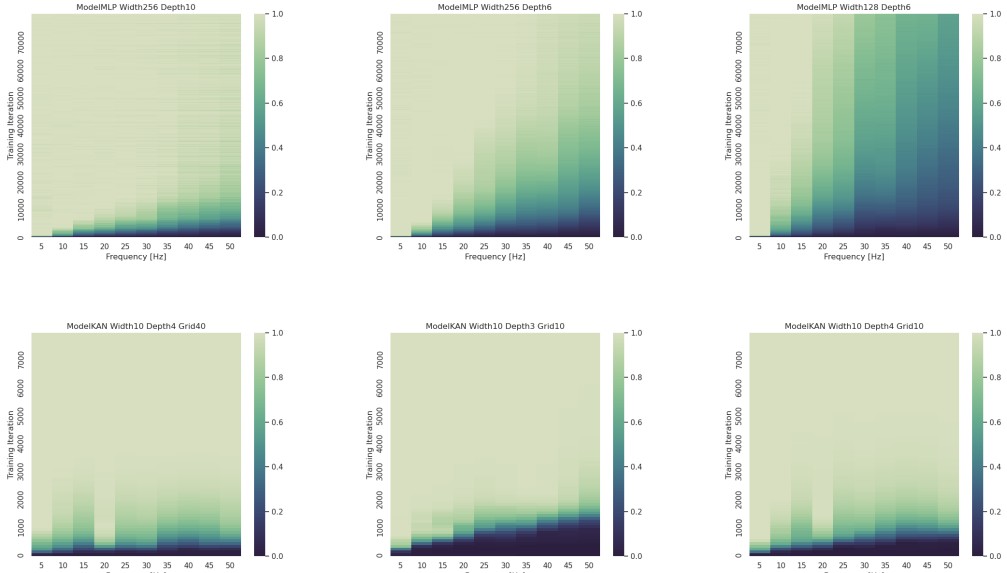

Figure 1: 1D wave dataset, where the target function has equal amplitudes of different frequency modes. Under various hyperparameters, MLPs manifest strong spectral biases (top), while KANs do not (bottom). Note that the y axis (training steps) of MLP is 10 times that of KAN.

where $\xi_i$ are i.i.d standard Gaussians $N(0, 1)$. For $f$ with different scales $\sigma$ and dimensions $d$, we split the points into $80\%$ training and $20\%$ testing points. We use MLPs and KANs with different sizes to regress on the training set, with the mean squared loss as the loss function. For MLPs, we use $500$ iterations of LBFGS iteration, and for KANs, we use the grid extension technique, with grid sizes $(10, 20, 30, 40, 50)$, each trained with $100$ iterations of LBFGS. We can see that ReLU$^k$ MLPs performs even worse than a standard MLP.

We plot the loss curves here and compare the losses of different scales $\sigma$ and dimensions $d$, using an MLP of 256 neurons in each hidden layer and optimized over its depth, and KANs with 10 neurons in each hidden layer and $2, 3, 4$ layers; see Figure 2 for the regression loss on the training set with dimensions $2, 3, 4$ and scales $2^i$, $i = 0, -1, -2, -3$. We see that for larger scale and smoother functions, MLP performs better, while for smaller scale and rough functions, KANs perform better without suffering much from spectral biases, and grid extension is especially helpful. We remark that one can choose bigger grid sizes of KANs for smoother functions and obtain more accurate regressions.

Precisely since KANs are not susceptible to spectral biases, they are likely to overfit to noises. As a consequence, we notice that KANs are more subject to overfitting on the training data regression when the task is very complicated; see the second line of Figure 3. On the other hand, we can increase the number of training points to alleviate the overfitting; see the last line of Figure 3 where we increased the number of training and test samples by 10x. We remark that the current implementation of grid extension is prone to oscillation after refining grids during the undersampled regime, as observed in Rigas et al. (2024), and we will improve it in future works.

### 4.4 PDE EXAMPLE

In this example, we solve the 1D Poisson equation with a high-frequency solution, similar to Xu et al. (2019a). To be precise, consider the equation with zero Dirichlet boundary condition

$$-u_{xx} = f \quad \text{in} [-1, 1], \quad u(-1) = u(1) = 0. \tag{13}$$

Here for a frequency $k \in \mathbb{N}$, the right-hand-side and the associated true solution are

$$f = \pi^2 \sin(\pi x) + \pi^2 k \sin(k \pi x), \quad u = \sin(\pi x) + \frac{1}{k} \sin(k \pi x).$$

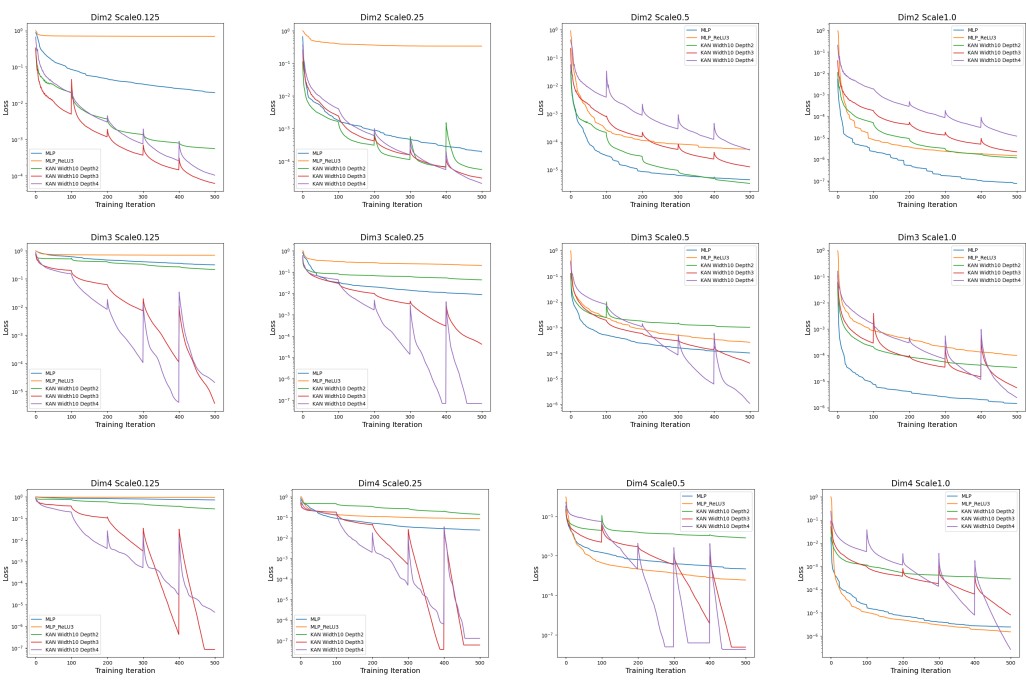

Figure 2: The Gaussian random field dataset. Training losses of MLP and KANs, with different scales and dimensions.

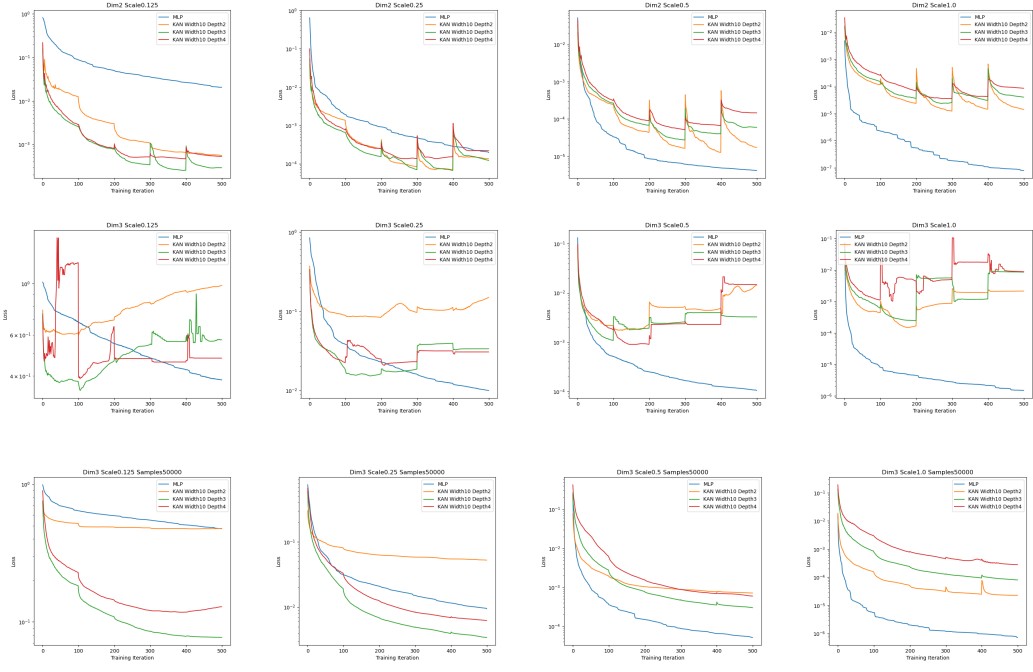

Figure 3: The Gaussian random field dataset. Test losses of MLP and KANs, with different scales and dimensions. Increasing the number of samples by 10x helps overfitting.

The different frequencies are normalized in a way that for $k > 1$, the ground truth has the same energy ($H^1$) norm. We use the variational form of the elliptic equation and the associated Deep Ritz Method Yu et al. (2018). Parametrizing $u$ by a neural network, we minimize the loss

$$\lambda \int_{-1}^{1} \left( \frac{1}{2} u_x^2 - fu \right) dx + u^2(-1) + u^2(1).$$

For frequencies $k = 2, 4, 8, 16, 32$, we use 2000 uniformly spaced sample points and the neural network using an MLP of 6 layers and 256 neurons in each hidden layer and a KAN of 2 layers with 10 neurons in the hidden layer. We choose the hyperparameter $\lambda = 0.01$ balancing the energy and boundary loss and perform LBFGS iterations. For MLPs, we use 200 iterations, and for KANs, we use grid sizes $(20, 40)$, each trained with 100 iterations. We plot the relative $L^2$ and $H^1$ losses compared to the ground truth in Figure 4. We can see that KANs perform consistently better, and the residue barely deteriorates when the frequency increases, whereas it becomes extremely hard for MLPs to optimize when $k = 16, 32$. We refer a 2D case to the appendix A.

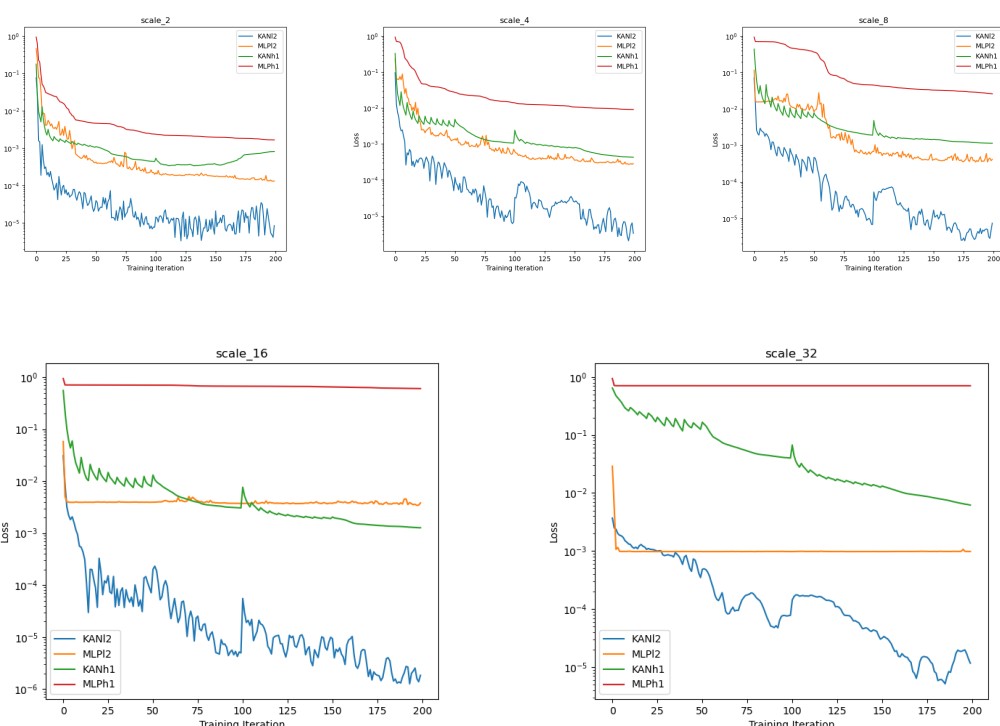

Figure 4: Solving PDEs. $L^2$ and $H^1$ losses of MLP and KAN with different frequencies of the solution.

## 5 CONCLUDING REMARKS

In this work, we have compared the approximation and representation properties of KANs and MLPs. Based upon our theoretical and empirical analysis, we conclude that while KANs and MLPs are very similar from the perspective of approximation theory, i.e. the both parameterize similar classes of functions, they differ significantly in terms of their training dynamics. Specifically, KANs do not exhibit the same spectral bias toward low frequencies that MLPs do. Future work includes developing theory which can describe the training of deeper KANs, performing a more extensive experimental investigation of the spectral bias of KANs on more complicated problems, and investigating whether the reduced spectral bias of KANs improves their performance on scientific computing applications such as solving PDEs.

## 6 ACKNOWLEDGEMENTS

The research of YW and TYH was in part supported by NSF Grant DMS- 2205590 and the Choi Family Gift Fund. JWS is supported by the National Science Foundation (DMS-2424305) as well as the Office of Naval Research (MURI ONR grant N00014-20-1-2787). We acknowledge Dr. Juncai He for identifying a minor mistake in the proof of Thm 3.3 and the referees' comments for improving the paper.

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

## A   2D POISSON EQUATION WITH HIGH FREQUENCY COMPONENTS

In this experiment, we solve a 2D Poisson equation with a high-frequency solution using the Deep Ritz method. The equation with zero Dirichlet boundary condition has the form

$$-u_{xx} - u_{yy} = f \ \text{ in} [0,1]^2, \quad u(x,0) = u(x,1) = u(0,y) = u(1,y) = 0. \tag{14}$$

with a right-hand side and a corresponding exact solution depending on the frequency $k$ as

$$f = 2\pi^2 \sin(\pi x)\sin(\pi y) + 2\pi^2 k \sin(k\pi x)\sin(k\pi y), \quad u = \sin(\pi x)\sin(\pi y) + \frac{1}{k}\sin(k\pi x)\sin(k\pi y).$$

For frequencies $k = 2, 4, 16$, we use a uniform grid with 101 sample points per direction with a MLP of 6 layers and 256 neurons in each hidden layer and a KAN of 2 layers with 10 neurons in the hidden layer. We use LBFGS for 50 iterations and plot the loss compared to the ground truth in Figure 5. We observe similarly as in the 1D case that KANs outform MLPs when solving equation with multiple scales.

More importantly, MLPs cannot learn the correct shape, even with $k = 4$ after 100 iterations, whereas KANs can learn a solution quite close to the ground truth with just 5 iterations for $k = 4$ and 10 iterations for $k = 16$; see Figure 6.

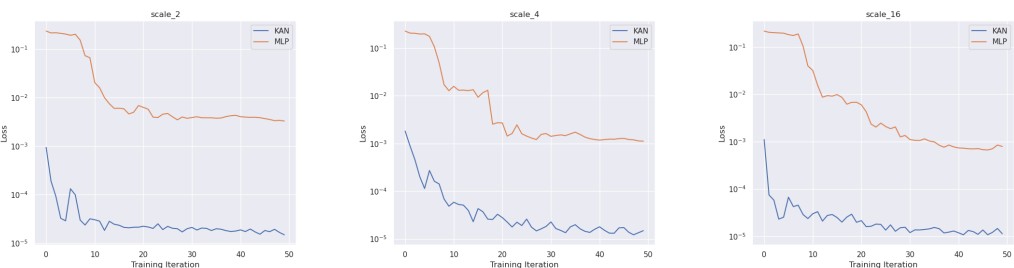

Figure 5: 2D Poisson. Losses of MLP and KAN with different frequencies of the solution.

## B   1D WAVES OF INCREASING AMPLITUDES

## C   PROOFS

*Proof of Theorem 3.2.* We will show that each layer of an MLP with the activation function $\sigma_k$ can be represented by a KAN with two hidden layers, width $W$ and grid size $G = 2$ with degree $k$ B-splines. By composing such layers, we obtain the desired result.

For a single layer of MLP, we consider the linear part and the non-linear activation separately. We first observe that on any compact subset of $\mathbb{R}^W$ the linear function

$$x_i = \sum_{j=0}^{W} a_{ij} x_j^{in} + b_i \tag{15}$$

can be represented with a single KAN layer of width $W$ by setting $\phi_{ij}$ to the linear function

$$\phi_{i,j}(x) = a_{ij}x + \frac{b_i}{n}. \tag{16}$$

We claim that this linear function can be exactly represented on any interval $[-R, R]$ in the form (4). To do this, we first set $w_b = 0$ and choose the grid points for the B-splines to be

$$\{-(2k-1)R, -(2k-3)R, ..., -R, R, ..., (2k-3)R, (2k-1)R\}.$$

Note that based upon the KAN architecture, this corresponds to the extension of the uniform grid $t_0 = -R, t_1 = R$ which has grid size $G = 1$. It is also easy to verify that there are $(k+1)$ B-splines supported on this grid, whose restriction to $[-R, R]$ span the space of polynomials of degree $k$. Thus,

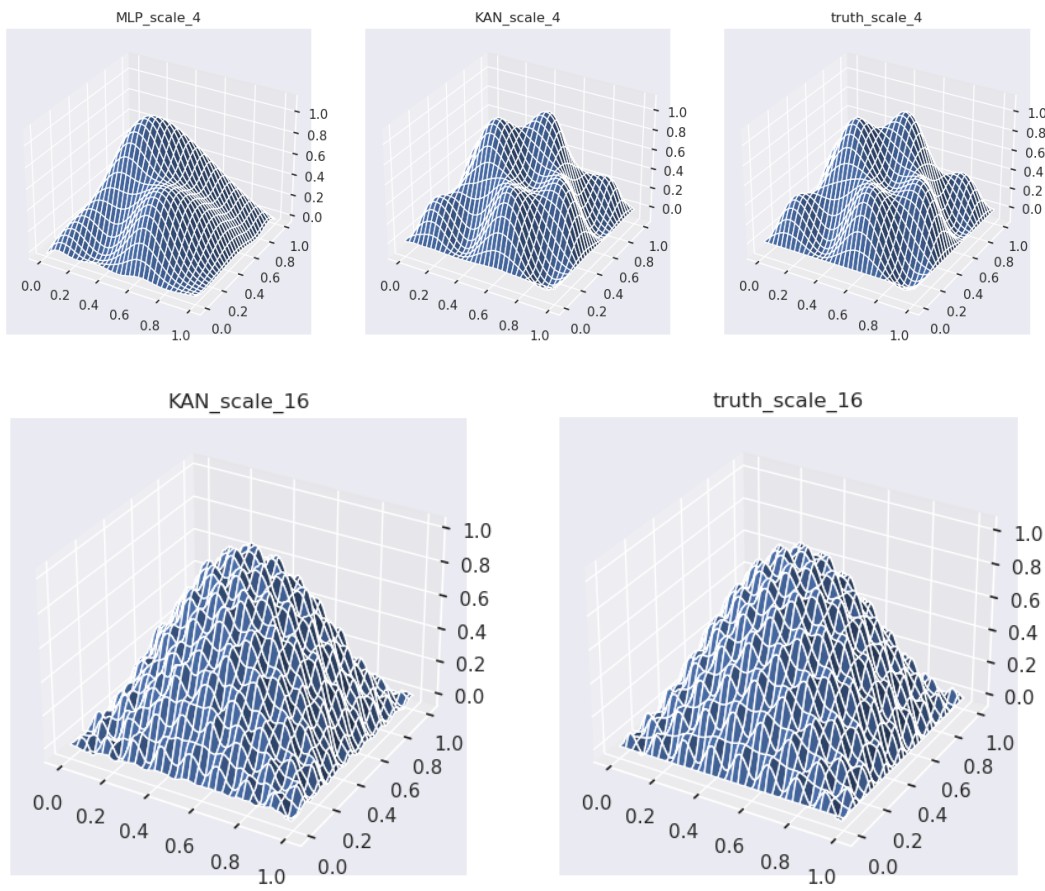

Figure 6: 2D Poisson. True solution and neural net solutions.

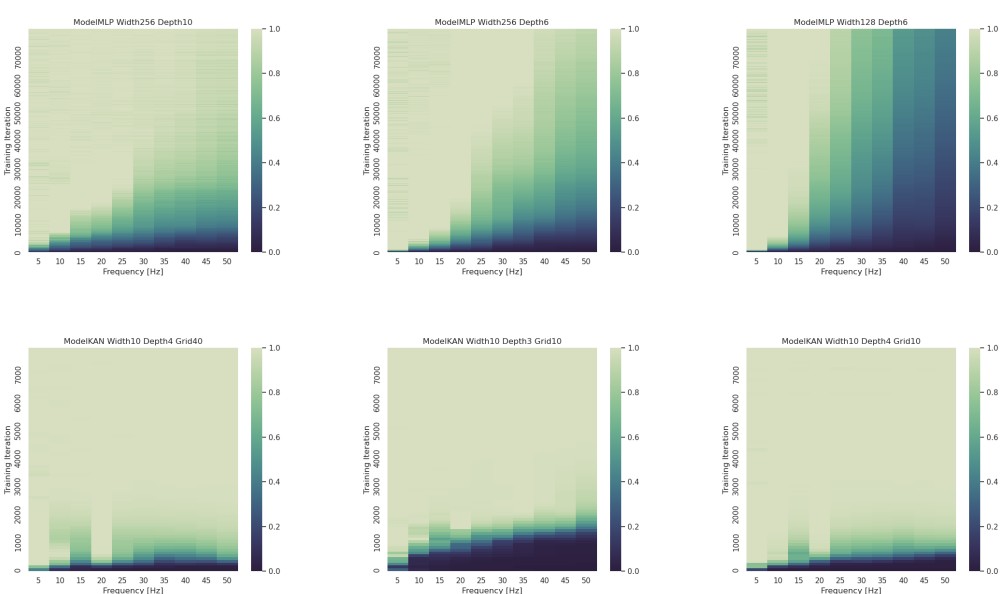

Figure 7: 1D wave dataset, where the target function has increasing amplitudes of different frequency modes. Under various hyperparameters, MLPs manifest severe spectral biases (top), while KANs do not (bottom). Note that the y axis (training steps) of MLP is 10 times that of KAN.

in particular, any linear function on $[-R, R]$ can be represented as a linear combination of these B-splines.

Next, we consider the non-linear activation, which is given by the coordinatewise application of $\sigma_k$, i.e.

$$x_i^{out} = \sigma(x_i). \tag{17}$$

This can be represented by a single hidden layer KAN by setting

$$\phi_{i,j}(x) = \begin{cases} \sigma_k(x) & i = j \\ 0 & i \neq j. \end{cases} \tag{18}$$

We claim that the functions $\sigma_k$ can be represented in the form (4) on any finite interval $[-R, R]$. To to this, we again set $w_b = 0$ and choose the grid points for the B-splines to be

$$\{-kR, -(k-1)R, ..., -R, 0, R, ..., (k-2)R, (k-1)R\}.$$

This grid is the grid extension of the uniform grid $t_0 = -R, t_1 = 0, t_2 = R$ which has grid size $G = 2$. It is easy also to verify that there are $(k+2)$ B-splines supported on this grid and that any piecewise polynomial on $[-R, R]$ with a single breakpoint at 0 which is $C^{k-1}$ is a linear combination of these B-splines. Hence the function $\sigma_k$ can be represented on $[-R, R]$ in the form (4) using this grid.

The proof is now completed by composing these layers and choosing $R$ sufficiently large so that for any input $x \in \Omega$ (which is bounded) the inputs and outputs of every neuron in the original MLP lie in the interval $[-R, R]$.

$\square$

*Proof of Theorem 3.3.* The proof follows from the fact that each KAN activation function $\phi_{l,i,j}$ can be represented by a ReLU$^k$ neural network with one hidden layer of width $G + 2k + 1$. This is due to the fact that each $\phi_{l,i,j}$ is a linear combination of $B$-splines (since we are assuming that $w_b = 0$ in (4)). This implies that it must be a compactly support piecewise polynomial spline of degree $k$ whose $(k-1)$-st derivative changes only on the extended grid which has $G + 2k + 1$ points. Based upon this, we can construct a shallow ReLU$^k$ network with one hidden node per grid point which matches each function $\phi_{l,i,j}$ (see for instance Xu (2020)). Since there are a total of $W^2$ such functions in each layer, this means that we can implement a single KAN layer using a shallow ReLU$^k$ network of width $(G + 2k + 1)W^2$. By composing these networks, we obtain the desired MLP representing the KAN. We remark that after grid extension, the grid points for each for each of the spline functions $\phi_{l,i,j}$ may become different. For this reason, it is not clear whether the width in this construction can be reduced. $\square$

*Proof of Theorem 4.1.* We first observe from (11) that the matrix $M$ is block diagonal with $d'$ identical blocks. Denoting these $(G+k-1)d \times (G+k-1)d$ blocks by $B$, it thus suffices to prove that

$$\frac{\lambda_{(G+k-1)d}(B)}{\lambda_d(B)} \leq C. \tag{19}$$

To do this, we analyze the blocks $B$ and note that they take the form

$$B = \begin{pmatrix} C & D & \cdots & D \\ D & C & \cdots & D \\ \vdots & \vdots & \ddots & \vdots \\ D & D & \cdots & C \end{pmatrix}. \tag{20}$$

Here the diagonal sub-blocks $C \in \mathbb{R}^{(G+k-1) \times (G+k-1)}$ are the Gram matrix of the one-dimensional B-spline basis, i.e.

$$C_{ij} = \int_0^1 B_i(x) B_j(x) dx, \tag{21}$$

and the off-diagonal sub-blocks $D \in \mathbb{R}^{(G+k-1) \times (G+k-1)}$ are rank one matrices

$$D = vv^T, \tag{22}$$

where the vector $v \in \mathbb{R}^{G+k-1}$ is given by

$$v_i = \int_0^1 B_i(x)dx. \tag{23}$$

It is well-known that the Gram matrix $C$ is well-conditioned uniformly in $G$ for a fixed $k$, i.e. $\lambda_{G+k-1}(C)/\lambda_1(C) \le K$ for a fixed constant $K$ depending only upon $k$. See for instance DeVore & Lorentz (1993), Theorem 4.2 in Chapter 5, where it is shown that the $L_2$-norm of a spline and the properly scaled $\ell_2$-norm of its B-spline coefficients are equivalent up to a constant depending only on $k$. This is equivalent to the well-conditioning of the Gram matrix $C$.

In addition, we can easily verify using Jensen's inequality (or Cauchy-Schwartz) that $D \preceq C$. Indeed, letting $w \in \mathbb{R}^{G+k-1}$ we see that

$$w^T D w = \left( \int_0^1 f(x)dx \right)^2 \le \int_0^1 f(x)^2 dx = w^T C w, \tag{24}$$

where the function $f(x) = \sum_{i=1}^{G+k-1} w_i B_i(x)$.

Let $\mathbf{1} \in \mathbb{R}^d$ be the vector of ones and note that

$$(v \otimes \mathbf{1})(v \otimes \mathbf{1})^T = \begin{pmatrix} D & D & \cdots & D \\ D & D & \cdots & D \\ \vdots & \vdots & \ddots & \vdots \\ D & D & \cdots & D \end{pmatrix} \tag{25}$$

so that $B - (v \otimes \mathbf{1})(v \otimes \mathbf{1})^T$ is a block diagonal matrix with diagonal blocks $C - D$. We proceed to upper bound the largest eigenvalue of $B$ by

$$\lambda_{(G+k-1)d}(B) = \max_{\|w\|=1} w^T B w$$

$$= \max_{\|w\|=1} w^T \begin{pmatrix} C - D & 0 & \cdots & 0 \\ 0 & C - D & \cdots & 0 \\ \vdots & \vdots & \ddots & \vdots \\ 0 & 0 & \cdots & C - D \end{pmatrix} w + (w^T(v \otimes \mathbf{1}))^2. \tag{26}$$

Writing $w = (w_1, ..., w_d)$ with $w_i \in \mathbb{R}^{G+k-1}$ and $\sum_{i=1}^{G+k-1} \|w_i\|^2 = 1$ and using that $D = vv^T$, we get the bound

$$\lambda_{(G+k-1)d}(B) \le \max_{\|w_1\|^2 + \cdots + \|w_d\|^2 = 1} \sum_{i=1}^d w_i^T C w_i + \left( \sum_{i=1}^d v^T w_i \right)^2 - \sum_{i=1}^d (v^T w_i)^2$$

$$\le \max_{\|w_1\|^2 + \cdots + \|w_d\|^2 = 1} \sum_{i=1}^d w_i^T C w_i + (d-1) \sum_{i=1}^d (v^T w_i)^2 \tag{27}$$

Since $D \preceq C$ we have $(v^T w_i)^2 \le w_i^T C w_i$ which gives the bound

$$\lambda_{(G+k-1)d}(B) \le d \max_{\|w_1\|^2 + \cdots + \|w_d\|^2 = 1} \sum_{i=1}^d w_i^T C w_i = d\lambda_{G+k-1}(C). \tag{28}$$

Next, we lower bound the $d$-th eigenvalue of $B$. For this, we use the Courant-Fisher minimax theorem to see that

$$\lambda_d(B) = \max_{W_d} \min_{w \in W_d, \|w\|=1} w^T B w, \tag{29}$$

where the maximum is taken over all subspaces of $W_d$ of codimension $< d$. We consider the specific subspace

$$W_d = \{(w_1, ..., w_d); \ v^T w_i = 0 \text{ for all } i = 1, ..., d\} \oplus \text{span}(v \otimes \mathbf{1}) \tag{30}$$

and observe that for any $(w_1, ..., w_d)$ with $v^T w_i = 0$ we have

$$w^T B w = \sum_{i=1}^d w_i^T C w_i \ge \lambda_1(C) \sum_{i=1}^d \|w_i\|^2 = \lambda_1(C)\|w\|^2, \tag{31}$$

while for the $w = v \otimes \mathbf{1}$ (which is orthogonal) we have

$$w^T B w = \sum_{i=1}^{d} v^T C v + (d-1) \sum_{i=1}^{d} \|v\|^2 \geq \lambda_1(C) d \|v\|^2 = \lambda_1(C) \|w\|^2. \tag{32}$$

Thus, $\lambda_d(B) \geq \lambda_1(W)$. Combining these bounds and using the well-conditioning of the Gram matrix $C$, we get

$$\frac{\lambda_{(G+k-1)d}(B)}{\lambda_d(B)} \leq d \frac{\lambda_{G+k-1}(C)}{\lambda_1(C)} \leq K \tag{33}$$

for a constant $K$ which only depends upon $k$. This completes the proof.

$\square$

