# OpenReview forum: "On the expressiveness and spectral bias of KANs"
_ICLR.cc/2025/Conference — ICLR 2025 Poster_

### Official Review · Reviewer_T7tm · 2024-10-31

**Soundness:** 3
**Presentation:** 2
**Contribution:** 2
**Rating:** 6
**Confidence:** 2

**Summary:**

The paper addresses the expressivity Kolmogorov-Arnold Networks (KAN) and the spectral bias.

A rigorous analysis of the representation power of the KAN vs MLP is welcome.

I think the most important contribution of the paper is to show the relationship between MLP with ReLU^k activation function and KAN without bias. Th.3.2 and Th.3.3 bounds are not tight. A conclusion of the authors is that MLP scales with G^2 while KAN with G, the grid size. The additional contribution Cor3.4 is not well explained in relationship to MLP, but the authors highlight the potential advantage of KANs.

The Analysis of the spectral bias seems a repetition of the analysis provided in [1], even if the authors analyze the Hessian matrix instead of the NTK matrix and show the elements of the Hessian. The authors are able to justify the reason why low frequency are not prioritized as in MLP, even if the connection with Th.4.1 is not clear.

The analysis of PDE, seems to be the same are reported in [1], showing better performance of KAN-based models in the training loss. Not sure why the test loss is not reported.


[1] 1. Wang, Y. et al. Kolmogorov Arnold Informed neural network: A physics-informed deep learning framework for solving forward and inverse problems based on Kolmogorov Arnold Networks. Preprint at https://doi.org/10.48550/arXiv.2406.11045 (2024).

**Strengths:**

As described above:

1. find the connection between two specific classes of MLP and KAN.
2. interpretation of the spectral bias using the Hessian matrix
3. some experiments to visualize the spectral bias

**Weaknesses:**

As reported in the summary:
1. the theoretical analysis is limited to specific class of MLP and KAN
2. experiments seem very similar to [1], so it is not clear, apart from Fig.1 and Fig.2, why they are included.

**Questions:**

1. could you expand on the consequences of Corollary 3.4
2. would be interesting to understand the consequence of thr.3.2 and th.3.3 for more "mainstream" architecture. For example, relu^k k=1 is probably the most common, while when gradients are needed SiLU SiLU-based activation functions are required, which is typical for scientific applications
3. would be nice to expand the Hessian analysis and connect with the NTK one
4. would be possible to perform the spectral analysis for section 4.4, instead of the error trend, if not, what is the test error?

**Details Of Ethics Concerns:**

the paper's scope is a generic computation architecture. no ethical aspects are relevant here.

---

> ### Author Response · Authors · 2024-11-19
>
> 1. Analysis of the spectral bias and PDE example very similar to [1]:
>
> Thanks for the comment and sorry that we did not notice earlier. We agree that [1] used NTK to empirically analyze the spectral bias and perform an example very similar to our PDE example. However, notice that they only used that example for function fitting, not for PDE solving. Such an example is very natural and this paper proposed it unaware of [1]. NTK is only valid in the infinite width or lazy regime where the parameters don't change much, so the eigenvalues of NTK might not be the most indicative in the active learning regime during training. In contrast, our spectral bias analysis is focused on the finite width regime. Nonetheless, it was our oversight that we did not recognize their contribution in this aspect and we have modified it in the Prior Works section. We also believe that the NTK analysis will be helpful in future work analyzing deeper KANs.
>
> 2. Expand on the consequences of Corollary 3.4:
>
> This result on general Sobolev spaces complements Thm 3.1 in the more restrictive class, in terms of approximation theory. It shows that for the general class, KANs are at least as good as MLPs (ReLU-k) for approximation.
>
> 3. Understand the consequence of thr.3.2 and th.3.3 for more "mainstream" architecture like ReLU and SiLU:
>
> Thanks. In terms of exact representation, it might be hard since the nonlinearities are different from that of B-splines. However, we invoke Thm 3.2 mainly to derive Cor 3.4, where we achieve a very nice approximation rate in the general Sobolev spaces. This complements Thm 3.1 in the more restrictive class, in terms of approximation theory.
>
> 4. Expand the Hessian analysis and connect with the NTK one:
>
> Thanks for the suggestion. As explained in point 1, NTK is only valid in the infinite width or lazy regime where the parameters don't change much, so the eigenvalues of NTK might not be the most indicative in the active learning regime during training. We are concerned with a finite number of neurons, where NTK regime is not valid even in the shallow case. The Hessian analysis drives the exact dynamics of training, while it is only valid in the shallow case. Notice that the trace of the Hessian, when output dimension $d'=1$, equals the trace of NTK as the number of points goes to infinity. We have added additional remarks comparing these two regimes.
>
> 5.  Perform the spectral analysis for section 4.4, instead of the error trend, if not, what is the test error:
>
> Thanks. We are in the regime of solving PDEs instead of function fitting, and we report the $L^2$ and $H^1$ losses rather than the training loss, which is the equation residue.

---

> > ### Comment · Reviewer_T7tm · 2024-11-26
> > **thank you for the reply**
> >
> > Dear authors,
> >
> > thank you for your replies.
> >
> > I would keep my score,
> >
> > Thank you

---

### Official Review · Reviewer_Wf2h · 2024-10-31

**Soundness:** 3
**Presentation:** 3
**Contribution:** 3
**Rating:** 6
**Confidence:** 2

**Summary:**

This paper is both a theoretical and experimental study of KANs, a recently introduced architecture with parametrizable activation functions,
aiming at characterizing their expressivity and spectral biases in comparison to MLPs. Since KANs offer a lot of new potentialities, it is
a legitimate and relevant research subject to try to characterize these  in  relation to properties of the data, therefore expressivity and spectral biases constitute obviously a good angle of analysis.

So the first theoretical result concerns expressivness: it tells that an arbitrary MLP (with activation function in C^k typically) can be exactly mapped onto a KAN (with B-splines of order k) with a depth multiplied by 2 and requiering a grid size of 2, so overall the number of parameters is multiplied by 4 (if I interpret well the result, this could be stated more explicitly actually) in this mapping. In reverse they find a mapping of a KAN to an MLP with an increase of parameters of G, the grid size of the original KAN. These constructions lead the authors to conclude qualitatively that KAN are more expressive than MLP. As acknowledged by the authors, this is not a sharp result as it is not excluded that a better construction of the mappings could be found leading to different conclusions, but these constructions are interesting on their own. A more concrete statement on expressivity is given by corollary 3.4, giving the speed of approximating of any function on a Sobolev space w.r.t the number of parameters of a KAN. Here, as a non specialist of this kind of result I become slightly confused. The statement  is seemingly based on generic results on MLP, and seems not to be specific to KAN but more to multi-layer models, as it exploit the  aforementioned mapping of the KAN to MLP, so that I wonder why MLP with relu-k would not lead to the same statement?
There is sentence discussing this at the end of the paragraph which is cryptic to me as it refers to some apparently non-standard MLP with relu-k.  As a result the overall take-home message of this part is unclear to me. Is there any advantage to be attributed to KAN wrt to comparable MLP under this measure of expressiveness? Maybe the statement 3.4 is a profound and central statement of the paper, but the discussion should make it clear, by first giving how standard MLP with relu-k compare and then also maybe what are the practical consequence of such statement if any?

The second theoretical point concerns spectral bias, and theorem 4.1 to me is a nice and non-trivial statement showing that a single layer of B-splines is well conditioned, especially when the grid size becomes large, by looking at the spectrum of the Hessian of the loss. This fully characterizes the conditioning of the problem with explicit dependency w.r.t. embedding dimension and grid size. The comparison is done with MLP with Relu using a result given in Hong et al. (2022) obtained with similar techniques (Courant Fisher theorem) showing  a much larger spectral bias for MLP. I wonder however if a  comparison with the spectral bias of MLP with relu-k (unfortunately not to given in Hong et al, but I presume that similar techniques would lead to the result) would not be more relevant as they are in the same class of regularity as KAN with B-splines of order k?

The experimental part focuses on the spectral bias question on diverse examples. A comparison is shown between KAN with kth order B-splines and MLP with relu, showing a clear advantage to KAN regarding spectral bias, but again I wonder why the comparison is not done with the spectral bias of MLP with relu-k, if this  would not be be more meaningful?

**Strengths:**

the questions addressed in the paper are interesting and some of the statements are quite strong, in particular th. 4.1 on the characterization of the spectral bias. Overall, my non-specialist viewpoint  on this paper is that it looks like a rather clear and solid paper, with interesting statements.

**Weaknesses:**

I found however a potential problem  of coherence concerning the comparison of KANs with MLPs: with relu-k at the beginning which is indeed legitimate,  and then basic relu starting from corollary 3.4.

**Questions:**

- Is the comparison between KANs with kth order B-splines with MLP modified when considering relu-k instead of simple relu?
- In Figure 3 and 4 the learning of KANs display instabilities in contrary to MLP which are attributed to grid extension during the undersampled regime, I don't really understand what is meant by this, is it that the grid size of KAN is progressively extended during learning?

Finally as a minor point:
I found that the way the experimental results are presented could be more concise and more informative. Figures 1 and 2 are almost identical, while all the plots given in figure 3 and 4 could be possibly synthetized to give the general behaviour w.r.t. to k, grid size, dimension and scale, and the legend should be magnified, I can't read anything on a printed version.

---

> ### Author Response · Authors · 2024-11-19
>
> 1. Elaborate on statement 3.4:
>
> Thanks for the question. As we mentioned after the statement of the theorem, this is the same rate in terms of approximation as MLPs in the Sobolev class. See also remark 3.5 for how this compares with the superior rate in Thm 3.1.
>
> 2. Comparison between KANs with kth order B-splines with ReLU-k MLP for spectral bias:
>
> Thanks for the valuable comment. We agree with the suggestion and have incorporated the comparisons. Basically theoretically by the works of Zhang et al. (2023) and via experiments, ReLU-k MLP suffers more from spectral bias, on top of being harder to train.
>
> 3. In Figure 3 and 4 the learning of KANs display instabilities in contrary to MLP which are attributed to grid extension during the undersampled regime, I don't really understand what is meant by this, is it that the grid size of KAN is progressively extended during learning:
>
> Thank you for the question. We have specified the grid sizes there and given a recap of grid extension in Sec 3.1. The instabilities at the grid extension points were caused by the suboptimal way that grid extension were implemented in pykan package. On the other hand,  they were caused by the fact that there was not enough training data; see the instabilities even when training on the coarsest grid.
>
> 4.  The experimental results presented could be more concise and more informative:
>
> Thanks. Figure 1 and 2 were both included to draw analogy to what was observed in the original spectral bias paper Rahaman et al. (2019). Figure 3 and 4 were used to demonstrate training and test losses, and discuss overfitting so that one can see clearly the two losses. We have magnified the legends and put Figure 2 in the appendix. We have also added additional discussion about the conclusions we draw from the experiments, and added additional experiments to support our claims to the appendix.

---

### Official Review · Reviewer_EXrE · 2024-11-02

**Soundness:** 3
**Presentation:** 3
**Contribution:** 3
**Rating:** 8
**Confidence:** 4

**Summary:**

This paper focuses on two aspects of KAN and MLP. The first issue is the approximation and representation capacility to functions of both networks, and the second issue is the spectral bias of approximation. Both issues are very important and attractive.

**Strengths:**

This paper characterizes the representation capacility of approximation of KAN and MLP, with a considerably rigorous mathematical analysis.

**Weaknesses:**

On the expressiveness, it is merely a theoretical interest, and the proof of mathematical analysis is over simplified by assumpting extremely shallow KAN network, which violates the practical deep network situation, and can not be regarded as equalivent to general case or nonlinear effect.

The experiments are not sufficient, and the paper is mainly of theoretical approach.

**Questions:**

I have following questions:

(1).  All the theorems 3.1, 3.2 and 3.3 are extremely mathematical, and their assumptions are not agreeable with practical cases. How can the authors explain the main differences?

(2). For the examples of PDE, the equation is very simplie, and the functions are of sufficient smoothness. If the functions concerned becomes chirp function and so on, how about the performance?

(3) In Fig. 4, the performance of approximation does not always tend to better. Can you explain the causes?

---

> ### Author Response · Authors · 2024-11-19
>
> 1. All the theorems 3.1, 3.2 and 3.3 are extremely mathematical, and their assumptions are not agreeable with practical cases; and the proof of mathematical analysis is oversimplified by assumpting extremely shallow KAN network:
>
> Thanks a lot for the comment. The assumption of Thm 3.1 is mathematical but also relevant in lots of tasks including symbolic regression; see the original KAN paper Liu et al. (2024e). For Thm 3.2 and 3.3, we are comparing how one uses MLPs to represent KANs and vice versa. You probably also meant Thm 4.1 on the spectral bias; for expressiveness we don't assume shallowness. In terms of spectral bias, we have already acknowledged the restriction of our analysis and used the subsequent examples in various tasks to demonstrate the empirical observation for deep cases that KANs perform better for high-frequency tasks.
>
> 2. The experiments are not sufficient; For the examples of PDE, the equation is very simple, and the functions are of sufficient smoothness. If the functions concerned becomes chirp function and so on, how about the performance:
>
> Thanks. We agree with this comment. We use these examples mainly to demonstrate the spectral bias, and agree that these experimental results are preliminary. We have added some more challenging examples in the appendix, and leave a more extensive empirical testing of this phenomenon to future work.
>
> 3. In Fig. 4, the performance of approximation does not always tend to better:
>
> Yes indeed. For smooth tasks the MLPs behave better, but their performance degrades significantly for rough cases, which is exactly what we would expect based upon our analysis of the spectral bias. Also in the middle row, since we don't have enough training samples, KANs overfit, which is another point we are making showing that KANs don't suffer much from spectral biases as they fit the higher-frequency noises as well. We agree that our initial presentation was not clear and have added remarks clarifying these issues.

---

> > ### Comment · Reviewer_EXrE · 2024-11-27
> >
> > Thanks a lot for your replies. Actually, I believe that you've already recognized that there still exist many problems that must be explained. I am satisfied with your responses.

---

### Official Review · Reviewer_iJru · 2024-11-03

**Soundness:** 3
**Presentation:** 2
**Contribution:** 2
**Rating:** 5
**Confidence:** 5

**Summary:**

The paper addresses the theoretical and empirical properties of Kolmogorov-Arnold Networks (KANs), particularly focusing on their approximation capabilities and spectral bias. Given the growing interest in alternative neural network architectures for function approximation, the study of KANs is relevant to the broader machine learning and scientific computing community. However, while the comparison with MLPs is insightful, the direct impact and importance of KANs over existing methods are not convincingly demonstrated.

**Strengths:**

1.Theoretical Comparison of Expressiveness: The authors establish that KANs are at least as expressive as MLPs. Specifically, they show that any MLP can be reparameterized as a KAN with degree k splines which is only slightly larger. Conversely, they demonstrate that KANs can also be represented by MLPs, but the number of parameters increases significantly with the grid size of the KAN. This implies that KANs with large grid sizes may be more efficient in approximating certain classes of functions.
2. The exploration of KANs’ spectral bias and their comparison with MLPs provides a novel perspective. The use of B-splines and the grid extension method offer unique contributions. However, much of the theoretical groundwork relies heavily on established techniques.

**Weaknesses:**

1. The parameter count in KANs and MLPs are not directly comparable, and the authors have blurred the distinction between these concepts. Additionally, the depth of KANs differs from that of MLPs, and the authors should not directly compare them, as this leads to a conceptual confusion.
We acknowledge that the conceptual confusion arises from the discussion in Section 3, particularly in Theorem 3.1- 3.4, where we compare the parameter count and depth between KANs and MLPs. This issue is also present in the experimental results in Sec4.1 and 4.2, where parameter comparisons are made across different models.

2. Theorem 4.1 lacks clarity regarding its implications for KAN's performance on high-frequency data. The authors should explicitly state whether the theorem suggests that KANs outperform other models in handling high-frequency data. Additionally, it would be beneficial to provide a brief discussion or corollary connecting the theorem to spectral bias or high-frequency performance to clarify its practical significance.

3.The theoretical analysis provided in the paper is limited to shallow KANs, which restricts the generalizability of the conclusions regarding spectral bias. To enhance the practical relevance, the authors should discuss the limitations of their current analysis and suggest potential methods for extending the theoretical framework to deeper KANs. This would provide a more comprehensive understanding of the spectral bias in real-world applications.

4.While the experiments cover diverse applications, they lack rigorous statistical validation to substantiate the claimed advantages of KANs. The authors should include specific statistical tests or validation methods to strengthen their results. Furthermore, more detailed comparisons with state-of-the-art baseline methods would help provide a stronger foundation for the claimed benefits of KANs, especially in practical applications like PDE solving and Gaussian random field fitting.

5.Incomplete Problem Contextualization: The paper does not clearly establish the practical significance of overcoming spectral bias in real-world tasks, nor does it compare KANs against state-of-the-art methods beyond MLPs.

6.Limited Discussion on Limitations: The potential downsides of KANs, such as computational cost, overfitting, and practical implementation challenges, are not thoroughly addressed.

**Questions:**

1.Theoretical Expansion: How does the depth of Kolmogorov-Arnold Networks (KANs) influence their spectral bias and approximation capabilities? Could a deeper theoretical analysis be provided to explore this relationship?

2.Enhanced Experimental Design:Could the authors include more rigorous statistical analysis and comparative experiments with a broader set of baseline models, particularly state-of-the-art neural architectures used in scientific computing?

3.Real-World Applications: How could the practicality of KANs be demonstrated in more diverse and realistic scenarios? Are there plans to showcase examples that highlight their advantages and limitations?

4. Detailed Discussion: Could the authors provide a more detailed discussion on the computational cost, scalability, and limitations of KANs? What are the key factors that may affect their broader applicability?

5. The author should

---

> ### Author Response · Authors · 2024-11-19
>
> 1. The parameter count in KANs and MLPs are not directly comparable:
>
> We acknowledge that the parameters of KANs and MLPs are not directly comparable since they are different architectures. We have also added a remark that the number of parameters is not necessarily the most useful measure of complexity of a model. However, the total number of parameters is generally indicative of the efficiency of a model and as a first step in our analysis, we have taken this measure of complexity. We remark that future work would entail comparing KANs and MLPs using more fine-grained measures of complexity. On the other hand, we note that if one focuses on the running time; in the $3d$ GRF examples using $50000$ samples, KANs are almost ($\approx$74mins) as expensive as MLPs ($\approx$56mins) on a gpu, while in practice one can also make KANs even more efficient by optimizing the implementation. These additional remarks have been added in the appendix.
>
> 2. Implication of Thm 4.1 to spectral bias or high-frequency performance:
>
> Thanks for the suggestion. We have added more discussion about how the eigenvalues of the Hessian relate to the spectral bias, or performance of the model on high-frequency components. Specifically, Thm 4.1, compared with its MLP counterpart established in Hong et al. (2022) and Zhang et al. (2023), implies that for shallow models, KANs are able to fit high-frequency components of the target function more efficiently than MLPs when trained using gradient descent.
>
> 3. Restriction of the analysis to shallow KANs:
>
> We acknowledge that our analysis of the spectral bias is fairly restrictive (this is in contrast to our analysis of the expressiveness and approximation theory, which holds for general deep MLPs and KANs). We have added more clear remarks on the limitations of our analysis before section 4.2. Nonetheless, we argue that despite its limitations, our analysis provides a theoretical explanation for observation that KANs fit high-frequency data better than MLPs. Finally, we remark that in addition to the theoretical analysis, we also provide empirical evidence supporting this claim. We leave a detailed analysis of the spectral bias of deep KANs, which we expect to be quite challenging, to future work.
>
> 4. Statistical significance of the experiment:
>
> Thank you for pointing this out. We agree that in order to conclusively demonstrate a difference in the spectral bias of KANs and MLPs, one would require a large body of experiments covering a diverse set of practical applications., which is beyond the scope of a single paper. In our work, we have done a preliminary set of experiments that indicate a difference in the spectral bias of KANs vs MLPs. To the best of our knowledge, we have followed best practices and performed the relevant ablation studies. For instance, in the first example, we average over numerous random seeds. In the latter two experiments, we used a fixed random seed to make the experiments reproducible, but we have also observed a similar pattern using other random seeds. We have also added additional experiments to the appendix to strengthen our conclusions. In conclusion, we argue that our preliminary experiments are significant, and agree that future empirical work is necessary to conclusively validate our findings.
>
> 5. Theoretical Expansion: How does the depth of Kolmogorov-Arnold Networks (KANs) influence their spectral bias and approximation capabilities:
>
> Thanks for this question, which we believe is too complicated to include in this paper, and which we intend to leave for future work. For example, one could compute the eigenvalues of the NTK matrices of deeper KANs empirically to investigate their spectral bias from a theoretical perspective, see for example Wang et al. (2024). Deeper KANs have better approximation capabilities and rigorously studying their spectral bias properties is an important future research direction.
>
> 6. Limitation of KANs, including computational cost, overfitting, scalability,  practical implementation challenges, and real-world applications and comparison with SOTA models:
>
> This is a very good point, and in the paper, we remark on this limitation of our analysis. Indeed, we only take into account the approximation, representation power (in terms of the number of parameters), and spectral bias of KANs. Our analysis of the spectral bias of KANs is relevant to overfitting, and indeed our results indicate that KANs may be more prone to overfitting since they are more sensitive to higher-frequency noise. Other important practical considerations including those mentioned by the reviewer are not discussed in this work. We remark that some of these aspects are discussed in the contemporary literature, including some that we cite in the Prior Works section. Right now our observation is that KANs are able to address large-scale problems very well once we scale up the sizes of KANs, made possible by the more efficient implementations of KANs now.

---

> ### Comment · Reviewer_iJru · 2024-11-26
>
> Thank you!
> I have updated the score.

---

### Author Response · Authors · 2024-11-19
**Overall comment**

We thank all of the reviewers for their careful reading of the manuscript and the questions raised. We have tried our best to address the comments within the limited amount of time. We highlight the overall revisions we have made here based on the suggestions.
1. We have added a remark discussing the sharpness of theorem 3.3.
2. We have added more discussions on the implications of the theorems, especially 3.4 and 4.1.
3. We have added a more challenging PDE example in the appendix.
4. We have discussed the spectral bias and made comparisons with ReLU-k networks, both theoretically and experimentally.
5. We have added additional relevant references.

---

### Meta-Review · Area_Chair_t1ib · 2024-12-21

**Metareview:**

This paper studies KANs and presents a theoretical analysis of their approximation capabilities and spectral bias. The results suggest that, compared to MLPs, KANs provide a more compact representation when learning certain classes of functions. The analysis also shows that KANs are more sensitive to high-frequency components of the data, potentially making them more prone to overfitting. Considering the growing interest in exploring alternatives to conventional neural architectures, this paper provides a timely study and valuable insights. Reviewers appreciate the paper's contributions, highlighting the rigorous theoretical analysis and the interesting nature of the findings. They also value the visualizations and experiments that further support the theoretical claims.

However, reviewers also raised multiple concerns. Reviewer iJru identified several issues, including: an unfair comparison between KANs and MLPs in terms of parameter count and depth; unclear implications of the theory for high-frequency data; the limited scope of the theoretical analysis (also mentioned by T7tm), which focuses only on shallow KANs; experiments lacking comprehensive comparisons with state-of-the-art methods; and inadequate discussion of the limitations of KANs, such as computational cost and potential overfitting. The authors responded to these concerns, largely addressing the issues raised by the reviewer. The reviewer subsequently increased their initial score.

Reviewer EXrE expressed concerns that the "shallow model assumption" used to develop the mathematical analysis of KANs do not reflect practical deep network situations and cannot be considered equivalent to the general case or nonlinear effects. While acknowledging these limitations, the authors provided further context, which the reviewer found satisfactory. Reviewer Wf2h pointed out an issue with the comparison of KANs with MLPs, for which the authors provided an adequate explanation. Reviewer T7tm expressed concerns about the limited nature of the theoretical analysis (limited to shallow networks) and the similarity of the experiments to those in [1]. The authors acknowledged the similarity, admitting it was an oversight, but explained that [1] uses examples for function fitting, not PDE solving.

In summary, the majority of reviewers rated the paper in the "accept" zone (with one recommending a clear accept), and only one reviewer rated it as slightly below the acceptance threshold. Overall, the paper is slightly above the borderline, and given its strengths, as pointed out by the reviewers as well, and the importance of the problem studied, I believe it makes interesting and valuable contributions, and I recommend accept.

**Additional Comments On Reviewer Discussion:**

Reviewer iJru identified several issues, including: an unfair comparison between KANs and MLPs in terms of parameter count and depth; unclear implications of the theory for high-frequency data; the limited scope of the theoretical analysis (also mentioned by T7tm), which focuses only on shallow KANs; experiments lacking comprehensive comparisons with state-of-the-art methods; and inadequate discussion of the limitations of KANs, such as computational cost and potential overfitting. The authors responded to these concerns, largely addressing the issues raised by the reviewer. The reviewer subsequently increased their initial score.

Reviewer EXrE expressed concerns that the "shallow model assumption" used to develop the mathematical analysis of KANs do not reflect practical deep network situations and cannot be considered equivalent to the general case or nonlinear effects. While acknowledging these limitations, the authors provided further context, which the reviewer found satisfactory. Reviewer Wf2h pointed out an issue with the comparison of KANs with MLPs, for which the authors provided an adequate explanation. Reviewer T7tm expressed concerns about the limited nature of the theoretical analysis (limited to shallow networks) and the similarity of the experiments to those in [1]. The authors acknowledged the similarity, admitting it was an oversight, but explained that [1] uses examples for function fitting, not PDE solving.

---

### Decision · Program_Chairs · 2025-01-22

Accept (Poster)